Development and validation of a nomogram predicting multidrug-resistant tuberculosis risk in East China

He Fang 1 2
Wang Shu 3
Wang Hua 4
Ding Xing 5
Huang Pengfei 4
Fan Xiaoyun 1 xiaoyunfan@ahmu.edu.cn
1 Department of Geriatric Respiratory and Critical Care Medicine, The First Affiliated Hospital of Anhui Medical University , Hefei, Anhui , China
2 Department of Respiratory and Critical Care Medicine, Anhui Chest Hospital , Hefei, Anhui , China
3 Department of Geriatrics, The Third Affiliated Hospital of Anhui Medical University/Hefei First People’s Hospital , Hefei, Anhui , China
4 Department of Tuberculosis Diseases, Anhui Chest Hospital , Hefei, Anhui , China
5 Department of Tuberculosis Diseases, Anhui Provincial Hospital Infection District , Hefei, Anhui , China
Braga Erika
Electronic publication date: 2025 Feb 27
Publication date: 2025
Volume: 13
Electronic Location ID: e19112
Received 2024 Apr 11; Accepted 2025 Feb 13
Copyright: © 2025 He et al.
Copyright year: 2025
Copyright holder: He et al.
License: This is an open access article distributed under the terms of the Creative Commons Attribution License, which permits unrestricted use, distribution, reproduction and adaptation in any medium and for any purpose provided that it is properly attributed. For attribution, the original author(s), title, publication source (PeerJ) and either DOI or URL of the article must be cited.
License URL: https://creativecommons.org/licenses/by/4.0/

Keywords: Drug resistant pulmonary tuberculosis, Risk factors, Prediction, Nomogram, East China

Funding: Co-construction Project of Clinical and Preliminary Disciplines of Anhui Medical University 2021lcxk005, 2022lcxk020 Anhui Province Clinical Medical Research Transformation Special Project 202304295107020044 Anhui Provincial Health Research Program of China AHWJ2022b040 This work was supported by the Co-construction Project of Clinical and Preliminary Disciplines of Anhui Medical University (2021lcxk005, 2022lcxk020) and 2023 Anhui Province Clinical Medical Research Transformation Special Project (202304295107020044). The study was funded by Anhui Provincial Health Research Program of China (No. AHWJ2022b040). The funders had no role in study design, data collection and analysis, decision to publish, or preparation of the manuscript.

==============================
Objective

Multidrug-resistant tuberculosis (MDR-TB) is a global health threat. Our study aimed to develop and externally validate a nomogram to estimate the probability of MDR-TB in patients with TB.

Methods

A total of 453 patients with TB in Anhui Chest Hospital between January 2019 and December 2020 were included in the training cohort. In addition, 116 patients with TB from Anhui Provincial Hospital Infection District between January 2015 and November 2023 were included in the validation cohort. Multivariable logistic regression analysis was applied to build a predictive model by combining the feature selected in the least absolute shrinkage and selection operator regression model. The C-index, calibration plot, and decision curve analysis were implemented to evaluate the predictive model’s discrimination, calibration, and clinical practicality. Then, logistic regression and least absolute shrinkage and selection operator (LASSO) models were constructed using R software, and the accuracy, goodness of fit, and stability of the models were verified using the validation cohort.

Results

Eight variables of patients with TB were selected using the best penalization parameter of the LASSO regression method, and the nomogram was established. The model displayed good discrimination with a C-index of 0.752 and good calibration. A high C-index value of 0.825 could still be reached in the validation cohort. The decision curve analysis demonstrated the clinical value of the model.

Conclusion

In this study, we constructed the LASSO regression model based on eight clinical traits and outcomes of laboratory tests, providing a novel insight for evaluating MDR-TB.

Introduction

Tuberculosis (TB) is one of the oldest diseases known to affect human health and it is considered a global public health problem (Smith, 2003). TB remains a critical global public health threat (Farhat et al., 2024). In 2021, approximately 6.4 million new cases of TB and 1.6 million TB-related deaths were reported worldwide (Shang et al., 2024). Additionally, the 2022 World Health Organization (WHO) report indicates that around one-quarter of the global population is latently infected with Mycobacterium tuberculosis (Alsayed & Gunosewoyo, 2023; Bagcchi, 2023). Despite the improvement in the cure rate for TB in recent years, widespread transmission and increasing drug resistance continue to pose significant threats (Dheda et al., 2024; Farhat et al., 2024). According to WHO, TB continues to be a heavy burden in some developing countries (WHO, 2021). According to a 2019 WHO report, the three countries with the largest share of the global TB burden are India (27%), China (14%), and the Russian Federation (9%). Multidrug-resistant TB (MDR-TB) refers to TB that is at least resistant to the two most effective first-line anti-TB drugs isoniazid and rifampicin and also confirmed by a drug sensitivity test (Seung, Keshavjee & Rich, 2015). MDR-TB is considered a global public health problem and a major threat to the control and elimination of TB in the world.

However, only about one-third of patients diagnosed with MDR-TB are admitted to treatment each year. The MDR-TB therapy success rate is about 60%. The poor treatment results and limited access to MDR-TB treatment likely contribute to an increasing MDR-TB prevalence (Yan et al., 2023). Because of the long infection periods of MDR-TB, poor treatment effects, expensive treatment drugs, and severe side effects, the treatment of MDR-TB is not universally optimal, which hinders the prevention and control of TB (Gagneux, 2009; Pooran et al., 2013). If the treatment fails, these patients may once again become the source of infection of drug-resistant (DR) TB, thus seriously impacting social public health and economic development. Increasingly, researchers have reached a consensus on the importance of early prediction and assessment of MDR-TB risk, with the most important step being to identify individuals at risk of disease progression (Horton et al., 2023).

In recent years, to reduce the incidence of MDR-TB, some studies have aimed to identify the high-risk factors of MDR-TB patients by constructing prediction models. The nomogram has been identified as a practical tool for preventive intervention in clinical practice (Dupont, Blume & Smith, 2016; Horton et al., 2023; Wang et al., 2018). In addition, nomograms can also predict and evaluate the individualized risk of a disease, and quantitatively demonstrate the individualized probability of predicting the incidence of disease outcomes (Mulhall et al., 2019).

Previous studies demonstrated the value of symptoms such as fever and night sweats, imaging manifestations of multi-leaf lesions, and some laboratory parameters (serum TB antibody, T-cell Spot (T-SPOT) test, C-reactive protein and albumin (Alb)) in predicting TB activity and prognosis (Li et al., 2019; Nijiati et al., 2023).

In recent years, studies have proven that the interaction between the coagulation system and inflammation may ultimately be related to the increase in replication of Mycobacterium tuberculosis (Caccamo & Dieli, 2016). Therefore, the coagulation function of TB patients has attracted increasing attention. In our study, these laboratory parameters that may be related to treatment outcomes are included in the developmental scoring system. We have constructed an accurate and personalized but simple predictive nomogram in the hope of reducing the incidence rate of MDR-TB.

Materials and Methods

Ethics and patients

Research approval was granted by the Anhui Chest Hospital’s Ethics Committee (Approval No. K2023-013) and The First Affiliated Hospital of USTC (Anhui Provincial Hospital) Medical Research Ethics Committee (Approval No. 2024-RE-9). Following the Good Clinical Practice for Drugs in China (GCP), a waiver of the need for informed consent was obtained from the participants of the study. Therefore, no explicit consent was sought from the participants for their involvement in the study. For this retrospective study, we recruited patients as a training cohort from January 2019 to December 2020 at Anhui Chest Hospital, consisting of 229 MDR-TB cases caused by TB-causing bacteria resistant to two or more first-line anti-TB drugs, including at least isoniazid and rifampicin, and 224 non-MDR-TB cases.

Non-MDR-TB cases are patients with other forms of pulmonary TB except MDR-TB, including patients who are resistant to either isoniazid or rifampicin. The study included newly diagnosed patients, patients switching treatments, and patients with relapse or treatment failure. The resistance profile was determined using specific resistance testing methods, such as culture methods or molecular techniques. The high proportion of MDR-TB reflects the diversity of the patient population and the resistance patterns. The validation cohort included 116 TB cases from January 2015 to November 2023 at Anhui Provincial Hospital Infection District. Participants originated from Anhui Province and the surrounding East China region. Both the Anhui Chest Hospital and Anhui Provincial Hospital Infection District are specialized in treating infectious diseases in China. In this study, the patients were classified into “MDR-TB group (i.e., the MDR group)” and “non-MDR-TB group (i.e., the non-MDR group)” according to the outcome. The inclusion criteria were as follows: (1) For patients with pulmonary TB who received treatment at our hospital, the diagnosis of pulmonary TB referred to the “Diagnostic Standards for Pulmonary Tuberculosis” (Bigio et al., 2021); (2) The medical record information was complete; (3) Both cognitive functions and mental state were normal; (4) Could be followed up. Exclusion criteria were as follows: (1) The patient was lost to follow-up and could not be contacted or missing information; (2) Patients with changes in diagnosis during the treatment; (3) Patients with severe cognitive disorders or serious physical constraints.

Data collection

Baseline medical data of each patient was extracted through the hospital medical record system as follows: (1) Sociodemographic data included age, gender, marital status, areas of residence, a history of direct contact, education level, family income, and smoking history; (2) Clinical data included laboratory indexes blood routine, coagulation, liver and kidney function, sputum acid-fast bacillus smear, and imaging examinations; (3) Symptoms related to TB include cough, sputum production, fever, night sweats, hemoptysis, weight loss, chest pain, diarrhea, and others; (4) Whether invasive surgery was performed during hospitalization, including bronchoscopy, closed thoracic drainage, mechanical ventilation, indwelling catheters, gastroscopy and indwelling gastric tube, and deep venous catheters, and others; (5) Accompanied by chronic diseases, such as hypertension, diabetes, coronary atherosclerotic heart disease, hepatitis B, tumor, connective tissue disease, kidney disease and autoimmune disease; (6) Accompanied by extrapulmonary TB.

Development and validation of the risk model

Multivariable logistic regression analysis was conducted to build a predictive model incorporating the features selected using the least absolute shrinkage and selection operator (LASSO) regression method (Li, Lu & Yin, 2022). The LASSO regression was conducted using the “glmnet” package. The cross-validation method was used to obtain the expenditure parameter (lambda) corresponding to the maximum area under the curve (AUC) value. The discrimination, calibration, and clinical usefulness of the predictive model were assessed using the C-index, calibration plot, and decision curve analysis, respectively. A nomogram prediction model was constructed using R software. The accuracy, goodness of fit, and profitability of the model were verified. External validation was performed using the bootstrapping method, which was employed to estimate the standard error and assess the stability and reliability of the model’s results.

Statistical analysis

Descriptive statistics summarized the study population’s baseline characteristics. Continuous variables were presented as mean ± standard deviation or median (interquartile range), while categorical variables were presented as frequencies and percentages. Univariate logistic regression analysis identified potential risk factors associated with MDR-TB using a significance level of p < 0.05. These significant variables were included in the multivariate logistic regression model to determine independent predictors of MDR-TB, calculating odds ratios (ORs) and 95% confidence intervals (CIs) (Wang et al., 2018). The predictive model’s discrimination was evaluated using the C-index, while calibration plots assessed its calibration. Clinical usefulness was assessed through decision curve analysis (Vickers & Elkin, 2006). External validation was conducted using data from Anhui Provincial Hospital Infection District. All statistical analyses were performed using R software version 4.0.3, considering p < 0.05 as statistically significant.

Results

Differences in clinical features between MDR-TB and TB

Based on clinical characteristics and treatment response of MDR-TB, we selected 10 clinical feature factors (creatinine (CREA), fever, fibrinogen (FIB), invasive operation, disease duration, lymphocyte (LYMPH), age, platelet (PLT), D-Dimer (DD), activated partial thromboplastin time (APTT)) to analyze potential differences between MDR-TB and TB groups. Specifically, through a comparison between MDR-TB and TB, statistically significant differences were observed in these 10 clinical feature factors between the two groups (Fig. 1). In addition, compared with the normal group, the number of PLT, contents of DD and FIB, age, percentage of invasive operations, and fever were lower in patients with MDR-TB. APTT, CREA, course of disease, and lymphocyte count were higher in these patients. These differences might be related to the development, clinical manifestations, and treatment response of MDR-TB.

Figure 1 Different clinical features of patients with MDR-TB and normal TB.

Points represent values, p < 0.05 as statistically significant.

Development of an individualized prediction model

First, we conducted a multivariate logistic regression analysis to assess the relationship between these independent variables and the target variable. We calculated various indicators such as regression coefficient, standard error, odds ratio (95% CI), and p-value. The findings from the multivariable logistic regression analysis suggest that CERA (OR = 2.85, 95% CI [1.852–4.405], p = 0.00), Fever (OR = 0.44, 95% CI [0.255–0.759], p = 0.00), Invasive operation (OR = 0.42, 95% CI [0.261–0.671], p = 0.00), Disease duration (OR = 1.00, 95% CI [1.001–1.007], p = 0.00), Age (OR = 0.99, 95% CI [0.972–0.998], p = 0.02), and DD (OR = 0.55, 95% CI [0.314–0.956], p = 0.04) are independent predictors of MDR-TB (Table 1). We used the LASSO regression method to further select variables and build a risk model for MDR-TB. Figure 2 demonstrates the feature selection process of LASSO. Considering the limited contribution of disease duration and PLT to the model’s performance due to their small effect sizes, these variables were excluded. However, we acknowledge that effect size may be scale-dependent, and small effects might still play a role in prediction. The remaining eight factors were included in the LASSO model. Figure 2 demonstrates the optimal lambda value and shows the preservation of 8 significant variables at this value. These significant variables had non-zero coefficients in the LASSO regression model. This indicated that these eight clinical features might help evaluate MDR-TB.

Table 1 Multivariate logistic regression analysis of MDR-TB.

Susceptible factors	Regression coefficient	Standard error	Odds ratio (95% CI)	P	
CREA (μmol/L)	1.05	0.22	2.85 [1.852–4.405]	0.00	
Fever	−0.81	0.28	0.44 [0.255–0.759]	0.00	
FIB (g/L)	−0.15	0.09	0.86 [0.714–1.036]	0.11	
Invasive operation	−0.87	0.24	0.42 [0.261–0.671]	0.00	
Disease duration (months)	0.00	0.00	1.00 [1.001–1.007]	0.00	
LYMPH (*109/L)	0.38	0.25	1.46 [0.904–2.383]	0.12	
Age	−0.02	0.01	0.99 [0.972–0.998]	0.02	
PLT (*109/L)	0.00	0.00	1.00 [0.996–1.001]	0.24	
DD (mg/L)	−0.60	0.28	0.55 [0.314–0.956]	0.04	
APTT (sec)	0.08	0.04	1.09 [1–1.183]	0.05	

Figure 2 LASSO regression variable screening process: lambda optimal value and retained variable screening at the optimal value.

Model performance evaluation

To evaluate the performance of the established model on new data, we included TB patient data collected from Anhui Provincial Infection Hospital District as our validation cohort. The training cohort consisted of 229 cases of MDR-TB and 224 cases of non-MDR-TB. The validation cohort included 116 TB patients, 51 of whom were MDR-TB and 65 were non-MDR-TB. We obtained important performance indicators by plotting ROC curves for both the training and validation cohorts. In particular, the ROC curve results showed an AUC of 0.759 for the training cohort and 0.825 for the validation cohort (Fig. 3). These results showed that our model was effective in judging MDR-TB and non-MDR-TB.

Figure 3 The ROC curve of the risk model for MDR-TB in the train and test cohorts.

Nomogram construction for MDR-TB

Based on the aforementioned results, we constructed a nomogram for the training cohort. In Fig. 4, a bar-line chart illustrates the weights and points corresponding to the predictive scores of the eight aforementioned factors. This allows for an intuitive estimation of individual risk without relying on an actual formula. The nomogram illustrates the impact of each clinical factor on the MDR-TB outcome (indicated by the length of the line segment), and corresponding scores are marked on the respective axes. By mapping the values of each variable onto the axes and connecting the points associated with all values, we obtain the risk estimate for the patient. For example, if a patient has a total score of 517 based on these eight indicators, the estimated risk of developing MDR-TB is 0.859.

Figure 4 The nomogram of a risk model for MDR-TB.

*P < 0.05, **P < 0.01, ***P < 0.001.

Validation of the predictive model

Subsequently, in the calibration curve of the prediction model, we observed that the model had good predictive ability for drug resistance risk in patients with TB. Specifically, when the nomogram-predicted probability of MDR-TB was approximately between 0.41 and 0.5, it suggested no statistical departure from a perfect fit between the predicted and observed values (Fig. 5A). The model displayed good discrimination with a C-index of 0.752. A high C-index value of 0.825 could still be reached in the validation cohort. The decision curve showed that the threshold probability of MDR-TB in patients with TB was 80–90% based on the nomogram in this study (Fig. 5B). The application of this nomogram to predict MDR-TB might provide significantly more benefit than either the treat-all scheme or the treat-none scheme.

Figure 5 The calibration curve (A) and decision curve (B) of the risk model for MDR-TB.

Discussion

TB is a global public health threat. In 2014, the World Health Assembly set the goal of ending TB by 2035. One of the most significant obstacles to ending TB globally is the uncontrolled spread of MDR-TB. Currently, only 55% of patients with MDR-TB worldwide have received treatment successfully (WHO, 2021). It is worth noting that the mortality rate of patients with MDR-TB is worrying in countries with high epidemic rates, particularly where traditional treatment programs are used. The mortality rate of patients with MDR-TB and those with extensive MDR-TB is 40% and 60–70%, respectively (WHO, 2021). Many MDR-TB cases do not receive timely diagnosis and treatment in some parts of the world due to limited laboratory capacity for drug resistance testing, as well as the significant side effects, high costs, and long treatment durations of second-line anti-TB drugs. As a consequence, these cases often progress to more severe forms of drug-resistant TB (Abubakar et al., 2022; Pontali et al., 2019). Therefore, early detection and intervention of high-risk patients with MDR-TB can help avoid adverse treatment outcomes, thus contributing to global TB control.

Our study compared the social, demographic, symptomatic, and clinical characteristics of patients with and without MDR-TB, and found 10 major risk factors related to MDR-TB. The purpose of this study is to provide a practical and convenient clinical tool, where drug resistance testing is limited, to help doctors find, manage, and treat MDR-TB as soon as possible. We choose to use a nomogram to visualize the risk prediction model graphically. The main advantage of using a nomogram is the ability to individually estimate patient risk based on their characteristics, helping in decision-making (Balachandran et al., 2015). Drug resistance in TB is a complex and dynamic process that can evolve over time. In clinical practice, patients typically need to undergo treatment for a period before a complete evaluation of the effectiveness of drugs. The primary goal of this study was to develop a model that could assist in assessing drug resistance in patients with TB. This model can guide the initial treatment decision-making process by identifying current MDR-TB, which may help improve treatment outcomes and prevent further complications.

Previous studies have identified youth as an independent risk factor for MDR-TB (Workicho, Kassahun & Alemseged, 2017). One key factor contributing to this risk is treatment adherence (Brown & Bussell, 2011). Compared with older adults, young people often exhibit poorer compliance with TB treatment, increasing the likelihood of developing drug resistance (Snow et al., 2020). Our study supported this finding, further emphasizing that younger patients were more likely to experience treatment interruptions, irregular medication intake, or failure to complete the full course of therapy. These behaviors contributed significantly to the emergence of MDR-TB. This finding suggested a negative correlation of age with MDR-TB development, as older patients tended to adhere better to treatment protocols, reducing the likelihood of drug resistance. Therefore, improving treatment compliance among younger patients is crucial. Increased attention during follow-up care, including more intensive monitoring and support for young individuals, may help prevent the MDR-TB and improve overall treatment outcomes. These findings are consistent with our results, demonstrating a negative correlation between age and MDR-TB.

Thrombocytopenia is another common and serious side effect of TB treatment, especially prominent in patients with MDR-TB (Minardi et al., 2021). This condition may be induced by anti-TB drugs, particularly medications such as rifampin and pyrazinamide, which can trigger immune-mediated platelet destruction or suppress platelet production in the bone marrow (Lugito et al., 2023). Studies have shown that thrombocytopenia is not only a side effect of TB treatment but also closely associated with the development of MDR-TB and poor prognosis (Cheng et al., 2024). Patients with MDR-TB often have a long and complex treatment history, including drug treatment failures and multiple regimen adjustments, which can cause immune system dysregulation and increase the risk of thrombocytopenia (Dheda et al., 2024). Furthermore, thrombocytopenia in patients with TB is closely related to the chronic progression of the disease, immune responses, and drug toxicity. This condition may exacerbate treatment risks and is associated with higher mortality rates in patients with MDR-TB (Shah et al., 2024). Existing studies have shown that low platelet counts are strongly correlated with the risk of death in patients with TB, particularly among those with treatment failure or relapse (Chiang, Centis & Migliori, 2010). Therefore, thrombocytopenia may not only be a side effect of TB treatment but also serve as a potential prognostic indicator, signaling a higher risk of drug resistance and adverse outcomes.

The site of TB infection forms characteristic granulomas, which are the host cell—mediated immune response to the characteristic inflammation caused by Mycobacterium tuberculosis (MTB). These granulomas contain MTB bacteria within macrophages, pulmonary exudates rich in fibrin, lymphocytes, and multinucleated giant cells surrounding the edges of fibroblasts (Sheets et al., 2011). In 1945, M. Chase demonstrated that immunity against MTB could not be transferred to animals through immune serum, but rather through CD4 T lymphocytes. Activated T lymphocytes release cytokines such as interferon-γ (IFN-γ), activate other resting monocytes/macrophages, and upregulate the production of tumor necrosis factor, reactive oxygen species, and nitric oxide in macrophages. These actions lead to the formation of granulomas and the effective containment of MTB within them (Musvosvi et al., 2023). In our study, high lymphocyte levels were associated with a higher risk of acquiring MDR-TB, which may be related to the immune response in tuberculosis.

Serum CREA is an indicator of kidney function, and it is often used to examine if kidneys are functioning properly. Although elevated, creatinine may not directly cause TB resistance but can affect the metabolism and excretion of anti-TB drugs. Anti-TB drugs, such as isoniazid and rifampin, are mostly metabolized and excreted by the kidneys (Chung et al., 2022). Therefore, if kidney function is abnormal or serum CREA is elevated, it may have an impact on the clearance of anti-TB drugs. This can lead to an excessive accumulation of drugs in the body and an increased prevalence of multidrug resistance (Adeniji, Knoll & Loots, 2020). Our study discovered that serum CREA becomes a potential driver in the emergence of MDR-TB. Therefore, serum CREA monitoring is crucial in the treatment of TB resistance. To guarantee the medication’s safety and effectiveness, physicians might modify the dosage and frequency of anti-TB medications based on the patient’s serum CREA levels.

Invasive procedures, such as surgery, can trigger various physiological responses, including inflammation and activation of the immune system (Margraf et al., 2020; Yang, Velagapudi & Terrando, 2020). One common consequence of these processes is fever or an elevated body temperature (Cordeiro, Lima Silva & Mendes, 2020; Crompton, Crompton & Matzinger, 2019). The immune system responds to fever by enhancing its defenses against pathogens, including MTB, the bacteria causing TB (Ravesloot-Chávez, Van Dis & Stanley, 2021). This heightened immune activity strengthens the body’s ability to combat the infection, thus reducing the replication and spread of tuberculosis bacteria (Ravesloot-Chávez, Van Dis & Stanley, 2021). Additionally, the increased immune response can help limit the development of drug resistance in TB, particularly MDR-TB. Fever boosts the bactericidal activity of immune cells such as macrophages, thus reducing the likelihood of TB bacteria developing resistance to anti-TB drugs (Fatima, Bhaskar & Dwivedi, 2021; O’Garra et al., 2013). In contrast, individuals with immune deficiencies, such as those with HIV co-infection, are at a higher risk of both TB infection and the development of drug resistance due to their weakened immune responses (Vinay, Abul & Jon, 2015). These findings support the idea that immune system activation, triggered by fever and invasive surgery, may play a protective role in reducing the risk of MDR-TB by enhancing the body’s ability to control bacterial growth and prevent resistance. Thus, a hyperactive immune response, induced by physiological stressors such as fever, may serve as a critical factor in reducing the likelihood of developing MDR-TB.

APTT, DD, and FIB are common indicators of coagulation function, and there is no literature supporting a direct relationship between APTT, DD, and FIB. Our study shows that prolonged APTT and decreased DD and FIB content are adverse indicators driving MDR-TB. As a plasma protein, FIB is involved in the regulation and maintenance of the human immune system, and its decrease may regulate the progression of MDR-TB by inhibiting immune function. In addition, since abnormal activation or dysregulation of the immune system leads to disorders of the coagulation system, the clotting process in turn activates immune cells and regulates their function (Wilhelm et al., 2023). The abnormality of coagulation function indicators such as APTT and DD may indirectly regulate MDR-TB by influencing immune system function, which requires more research to confirm.

Our study had several limitations. First, the data might not fully represent all pulmonary tuberculosis cases in China, as untreated patients were excluded. The sample size was relatively small, and the prevalence of TB in other regions or countries remained unclear. Additionally, factors affecting TB resistance, such as genetic mutations, socioeconomic conditions, and improper medication management, were not considered. Clinical prediction models also lack individualization and dynamism because they rely on static data and fail to account for the changes in a patient’s condition over time or individual differences. In this study, some predictors (such as a history of tuberculosis) may serve as proxies for the severity of the disease. Although this does not affect the goal of prediction modeling, it does have certain limitations. Therefore, a broader evaluation of diverse populations with TB is necessary. We plan to expand the sample size to enhance the robustness and generalizability of the model in the future.

Conclusions

Our study showed that age, APTT, DD, fibrinogen, creatinine, lymphocyte, invasive operation, and fever were the possible risk factors associated with MDR-TB. This may be a useful supplement to the identification of MDR-TB at present, especially in patients with MDR-TB who cannot provide the test samples.

Supplemental Information

Supplemental Information 1 R script to calculate and evaluate the concordance index (C-index) to assess the discrimination ability of the predictive model for multidrug-resistant tuberculosis (MDR-TB).

It helps determine how well the model differentiates between patients with and without MDR-TB.

Supplemental Information 2 R script to construct a nomogram based on the predictive model.

The nomogram offers a visual representation of risk prediction for MDR-TB and allows for the calculation of individual patient risk scores using the selected predictors.

Supplemental Information 3 The input data required for calculating the C-index, performing Decision Curve Analysis (DCA), and constructing the nomogram model.

Includes necessary variables and patient data to evaluate and predict the risk of MDR-TB.

Supplemental Information 4 R script to implement the Least Absolute Shrinkage and Selection Operator (LASSO) regression for feature selection.

This helps identify the most important predictors of MDR-TB risk, reducing overfitting by selecting a subset of relevant variables for the predictive model.

Supplemental Information 5 R script to perform Decision Curve Analysis (DCA) to assess the clinical usefulness of the predictive model.

This calculates the net benefit at different threshold probabilities, comparing the model’s performance to alternative treatment strategies.

Supplemental Information 6 R script to generate a calibration plot to assess the agreement between predicted and observed outcomes.

This evaluates how well the predicted probabilities match the actual observed risk of MDR-TB, providing insights into the model’s calibration performance.

Supplemental Information 7 Original data.

448 cases of tuberculosis patients from Anhui Chest Hospital and 116 cases of MDR-TB patients from Anhui Provincial Hospital Infection District.

Supplemental Information 8 Testing data.

Supplemental Information 9 Training data.

Additional Information and Declarations

Competing Interests

The authors declare that they have no competing interests.

Author Contributions

Fang He conceived and designed the experiments, performed the experiments, analyzed the data, prepared figures and/or tables, authored or reviewed drafts of the article, and approved the final draft.

Shu Wang conceived and designed the experiments, performed the experiments, analyzed the data, prepared figures and/or tables, authored or reviewed drafts of the article, and approved the final draft.

Hua Wang performed the experiments, authored or reviewed drafts of the article, and approved the final draft.

Xing Ding performed the experiments, authored or reviewed drafts of the article, and approved the final draft.

Pengfei Huang performed the experiments, authored or reviewed drafts of the article, and approved the final draft.

Xiaoyun Fan conceived and designed the experiments, performed the experiments, analyzed the data, authored or reviewed drafts of the article, and approved the final draft.

Human Ethics

The following information was supplied relating to ethical approvals (i.e., approving body and any reference numbers):

Research approval was granted from Anhui Chest Hospital’s Ethics Committee (Approval No. K2023-013) and The First Affiliated Hospital of USTC (Anhui Provincial Hospital) Medical Research Ethics Committee’s Ethics Committee (Approval No. 2024-RE-9).

Data Availability

The following information was supplied regarding data availability:

The original data is available in the Supplemental Files.

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
