# Peer review of "Development and validation of a nomogram predicting multidrug-resistant tuberculosis risk in East China"

_PeerJ, doi:10.7717/peerj.19112_

## Round 0.1 · original submission · Major Revisions

The reviewers have provided detailed guidance. Please address their comments in a revision. In particular, the comments of R2 must be adequately addressed in any revision

Note that the comments from R1 are in their appended PDF.

·

Basic reporting

Overall written with good language, but need

Experimental design

attached separate

Validity of the findings

attached separate

Additional comments

attached separate

Reviewer 2 ·

Basic reporting

There are considerable language issues throughout the manuscript, likely due to the authors imperfect understanding of all the statistical concepts and terms in the English language. The intention is likely mostly correct, but the incorrect wording may confuse or mislead the reader, i.e., this is likely a language issue, and not a scientific issue.
Below, I give an overview for the abstract and some of the more relevant issues, but this issue is present throughout the manuscript.

Abstract comments:
• “Multidrug-resistant tuberculosis (MDR-TB) is a huge social health threat.”
The word “huge” is subjective, the word “social” is confusing and unnecessary, it is an ongoing and increasing threat to global health. This is phrased more correctly in the introduction.
• “to develop and externally validate a nomogram to estimate the risk factors for MDR-TB infections in patients with TB.”
This is not correct. The nomogram is a graphical calculation device, in essence a way to visualize a calculation that was especially useful before the advent of computers and hand-held devices. It’s not a model to estimate the risk factors, that is done by the logistic model and LASSO. The results of the models are subsequently implemented in the nomogram which can then be used to calculate / estimate the probability of MDR-TB in patients with TB, but not the risk factors for MDR-TB.
• “Then, a nomogram prediction model was constructed using the R software, and the accuracy, goodness of fit, and profitability of the model were verified”
Again, the nomogram itself is not a model, it’s an implementation of a model that is estimated by another procedure (e.g., logistic regression + LASSO)
• “profitability of the model” and “the high model profitability”
Model profitability is not a term that I am aware of, and as far as I know not a correct term for a model evaluation.
• “Ten variables of TB patients were selected through the LASSO regression method, and a from which a nomogram was built.”
This is again not correct. Ten variables were included in the LASSO, of those, the number selected depends on the chosen value of the penalization parameter. Additionally, part of the sentence seems to be missing.
• “External validation was assessed using the bootstrapping validation method.”
Bootstrapping is a statistical technique, not a validation method.
• “the interval validation”
“interval” must be “internal”
• “The model has been found to be useful in clinical practice and is clinically interpretable”
There has been no clinical validation or implementation research to support these statements. This cannot be concluded from a statistical validation alone.

Language: There is often no space between the dot at the end of the sentence and the start of the next sentence.

Line 41: “TB continues to cause considerable morbidity and mortality globally, while it remains the leading cause of adult death from an infectious disease worldwide, with more than 10 million people becoming newly sick each year. TB is leading cause of death in humans after HIV/AIDS.”
These statements are not correct: TB was the leading cause of adult death among infectious diseases until it got replaced by COVID. It is certainly not the leading cause of death in humans after HIV/AIDS and worldwide not even in the top 10 depending on which subgroup or classification is used (ranking behind, among others, coronary diseases, stroke and several cancers). Maybe the authors want to imply that it is the highest cause of death in people with HIV, but even this does not seem to be supported by recent literature either.

Line 51: “The poor treatment results and limited access to MDR-TB treatment have led to the global prevalence of MDR-TB.”
This sentence has no true meaning. All events that ever happened in the world have lead to the global prevalence of MDR-TB. You likely want to imply that poor treatment results and limited access to MDR-TB treatment likely contribute to an increasing MDR-TB prevalence.

Line 82: “as a train group”
“train” should be “training”, better words that “group” exist as well, such as “cohort” used later.

Line 133: “Furthermore, compared with the normal group, patients in the MDR-TB group had reduced PLT number, DD and FIB content, younger age, and a decrease in the percentage of invasive procedures and fever, while APTT was prolonged, CREA, disease duration, and lymphocytes increased.”
These are not causal or sequential effects, so the words “reduced”, “decrease” and “increased” are used incorrectly and should be replace by words that do not indicate a change such as “lower” and “higher”.

Experimental design

Line 128: It’s a bit unclear which variables were tested as only the significant ones are shown.
For some variables, it’s also unclear if they can truly be used as predictors for MDR-TB, as the predictor may only be observed after the outcome.

Line 143: “Remarkably, the 10 differential factors that were selected through the Lasso method closely corresponded with the 10 clinical features tentatively hypothesized in Figure 1. This suggests a strong correlation between these 10 clinical features and MDR-TB.”
With the given information, this statements make no sense. Why were those 10 clinical features selected for LASSO in the first place? The authors never state why and how they chose the predictors to be included in the LASSO model, but the figures clearly imply that it’s exactly the 10 variables with a significant result in the univariable logistic regression that were selected. Hence, it’s absolutely no surprise that when none are removed, that these are still the exact same variables and therefore does not suggest much.
Additionally, it is very clear that the actual estimates are strongly shrunk at the chosen penalty parameter, with some much closes to 0 than they originally were. In the multivariate regression model, several variables are not significant, suggesting that the conditional correlation between these variables and MDR-TB is not necessarily strong or providing no statistical evidence that it truly exists for some of the variables (which for prediction is not all that relevant anyway).

Line 154: “These findings are consistent with the results obtained from the Lasso regression method, further confirming the effectiveness of Lasso regression and providing specific information about the most strongly associated predictors for MDR-TB.”
As mentioned in my comments before, this statement is not supported by the results, the findings are not fully consistent, and are consistent with a conclusion that some predictors may not be very useful. Additionally, for disease duration, the distribution is extremely right skewed, so it’s not clear how much of the effect is potentially caused by a limited number of outliers.

It is not specified how the optimal penalty parameter (lambda) is selected.

Line 123: “R software version 4.0.3”: this is a rather old version (from 2020), and older versions tend to have some package dependency issues, which makes it harder for people to replicate the results.

Validity of the findings

I noticed a few strange results in the "duration" in the validation dataset (subsequent records showing 11.4, 11.4, 114, 11.4), and a few duplicates in the duration in the training dataset as well, which is strange.

The conclusions are not well supported by the analysis, mostly due to the considerable issues with the experimental design discussed above.

The model is only statistically validated, statements on clinical practice are not supported by this work.
Additionally, nomograms can indeed have potential to be useful in clinical practice, especially in low resource settings, but this one seems quite complicated and not useful in a point of care setting, with a simple implementation of the prediction model on a handheld device likely much easier and faster in practice, making me doubt its clinical utility.

Additional comments

This research feels like a missed opportunity. The research idea is valid, and the attempt to both create and validate a prediction model with the proposed methodology is fundamentally sound and has potential.

Despite that, the following major issues were mentioned before:
* There are some suspicious records in the data
* The chosen variables for selection in the later models are not well documented or supported
* The actual analysis beyond the technical part is poor and poorly described
* The nomogram itself is too complicated for easy clinical use and not clinically tested despite a claim on the contrary being made in the abstract,

This severely limits the added value of this work and warrants, at minimum, a major and lengthy revision of this work before a resubmission in any journal.

Reviewer 3 ·

Basic reporting

Minor comments:

1. Line 43: TB is the leading...
2. Line 44: ...heavy burden disease in some...
3. Line 46: You defined MDR-TB in Line 9 already
4. Line 55: ...a source of drug-resistant (DR) TB infection,...
5. Line 64: Do you mean in clinical settings or clinical trials?
6. Throughout the paper there are often no spaces after commas and full stops.

Experimental design

Are the input files for the R scripts missing or is it all in that one supplementary excel table? The column names and numbers of samples do not match the numbers in the manuscript. like this i was not able to replicate the analysis. It might be I overlooked some data, please point me in the right direction.

Validity of the findings

No comment

Additional comments

Multi drug resistant tuberculosis is a major threat to global health. Therefore estimating risk factors for MTB infections is an important goal in public health efforts to put this disease under better control. Developing a simple to use and understand nomogram is in my opinion a step in the right direction.

---

## Round 0.2 · Major Revisions

Reviewer 2 has acknowledged that your article has been improved from its previous version, however it is clear that the core limitations remain unaddressed.

The reviewer has taken care to clearly explain what the issues are and so we ask that you carefully consider what they are saying. At the *minimum*, you should extensively address the major methodological limitations of your approach, although the *ideal* would be that you are able to improve the methodology and description of your results.

·

Basic reporting

ok

Experimental design

ok

Validity of the findings

ok

Additional comments

The authors have properly addressed my concerns and questions!

Reviewer 2 ·

Basic reporting

The basic reporting has greatly improved compared to the previous version.
Some minor language issues remain but these should be able to be overcome in a next version.

Overall, the goal of the manuscript is clear and clearly described and, if done sufficiently well, would provide added value to the research community.

Experimental design

Two new major issues came to my attention:

* How can an C-statistic and ROC curve be constructed from a dataset that has only true positive MDR-TB cases (the validation dataset)? In the actual dataset in supplementary, some patients are identified as "0" (negative). This contradicts the text in the manuscript and needs to be clarified.

* How is the outcome (MDR-TB vs. "normal") exactly defined? The current message is vague and a bit conflicting, hinting that it could be either the status at baseline, the status at end of treatment, or - most likely - patients are only included when the status at baseline and at end of treatment are the same.
Additionally, it's not clear to me what "normal" means, RR-TB? RS-TB? DR-TB with only one resistance? The authors should use exact scientific language to define how the outcome was defined, and ideally include a table with the exact drug resistance profiles of the patients included.

Otherwise, the methodology remains solid, although the issues raised in my previous review have not been fully resolved (see the validity section for details).

Validity of the findings

The updates to the methodology are sadly not sufficient and do not address the underlying issues discussed in my previous review. Please find below some issues that would need to be resolved, or at minimum addressed as major limitations, in a next version.

* Removing 2 variables, but still having 8 in, 8 out, most of those 8 being severely shrunk at the chosen penalty parameter does not provide solid evidence of the clinical value of these predictors. It is not wrong, but the authors still overstate the value and resulting evidence from this research, especially when taking into account the comment of another reviewer that the sample size is not that large.

* The advantages of the nomogram remain overstated. As it looks, researchers would have to manually try to estimate the sum of 8 numbers that are imprecisely estimated on a scale (with 20 point intervals), which is not trivial for anyone and especially not clinicians in the field, and can easily lead to errors. While I will not deny that a nomogram potentially provides some added clinical value, this is currently overstated.
Additionally, there was still at least one mention of the nomogram being a model itself, which it is not (see comment in my previous review).

* For continuous variables, estimates have to be interpreted relative to the chosen unit. E.g., a variable coded in years will have an estimate that is 12 times larger than if that same variable would be coded in months. This makes absolute interpretation of the odds ratios hard, and also makes the interpretation of the LASSO plot (Figure 2 on the left) hard as the shrinkage of the different parameters cannot be easily evaluated for those with coefficients close to 0, and cannot be relatively compared between the different parameters.

* When discussing issues with the data, I gave an example of 11.4, 11.4, 114, 11.4. The issue is not only that they are similar, it's also that it's 11.4 vs. 114. The author's answer confirms that it's highly likely that the 114 is a typo. The boxplot of disease duration also shows multiple patients that have a disease duration of over 10 years, with one outlier of 40 years. This may be no longer relevant since this variable was excluded in this version of the manuscript (although it is still included in one of the figures), but it does highlight that typos may have been overlooked in the dataset.

Additional comments

Please remember, when working on the exact same dataset, a result from a second statistical analysis method can never confirm the result of a first statistical analysis method in the population.

As an academic example: Suppose I have a dataset with (purely due to chance) tall women and small men. I first measure the participants with a tape measure, and then with a stadiometer. The fact that the stadiometer's measurements are in agreement with the tape measure's results is NOT evidence that women are taller than men. Statistically speaking, your type-I error is still the same, no matter how many times and in how many ways you measure the men and women.

Nevertheless, this is exactly what you do with your LASSO results and your initial univariate analysis, you use two methods that in essence doing the same thing (estimating an effect size), additionally using the results of one method as inclusion for the other method. This is not wrong, but their agreement can never be seen as evidence of anything, is not remarkable, and does not suggest a strong correlation of the predictors with the outcome more than any individual model does.

Reviewer 3 ·

Basic reporting

no comment

Experimental design

no comment

Validity of the findings

no comment

Additional comments

no comment

---

## Round 0.3 · Major Revisions

There has been only minimal improvement in each revision and the authors have failed to meaningfully address the prior comments. If the next revision is not significantly revised to an acceptable level then this article will be Rejected.

Reviewer 2 ·

Basic reporting

"tuberculosis bacteria that are resistant to two or more first-line anti-tuberculosis drugs such as isoniazid and rifampicin"
Replace "such as" by "including at least" as you say in your rebuttal: "Multidrug-resistant TB (MDR-TB) refers to TB that is at least resistant to the two most effective first-line anti-TB drugs isoniazid and rifampicin and also confirmed by a drug sensitivity test(Seung et al. 2015)."
Current formulation "Such as" would wrongly imply that it could also be two other first line-line drugs.

"Remarkably, the 8 differential factors that were selected "
Remove the word remarkably, as mentioned in earlier reviews, this is not remarkable at all but rather what you may reasonably expect given the earlier analysis.

"The training cohort consisted of 229 cases of MDR-TB and 219 cases of TB, while the validation cohort consisted of 116 cases of MDR-TB. "
Still incorrect, see the comments in the previous review!
* For the validation cohort, the data file suggests it's 51 cases of MDR-TB and 65 cases of non-MDR.
* For the training cohort, it's not 219 cases of TB (MDR-TB is also TB), it's 219 cases of non-MDR.

"Subsequently, in the calibration curve of the non-adherence risk line chart, we observed that the model has good predictive ability for medication non-adherence risk in MDR-TB patients."
Where is the non-adherence suddenly coming from? Isn't the model predicting MDR-TB? There is no mention of adherence anywhere else, nor is it in the validation dataset.
Additionally, a paragraph later, it's referred to again as MDR for figure 5B while this graph also mentions nonadherence.
Finally, I don't really see the 70-90% on graph 5B, it would rather be 80%-90%?

Experimental design

Some issues, mostly addressed in previous reviews and the other sections.

Validity of the findings

Still many issues with the discussion with many paragraphs overstating results, not properly linked to the work done here, or being confusing.
A few examples:

* "Since nomograms are simple, convenient, and at no cost, the application is easy to use and promote in clinical practice. In addition, the predictive capacity for each of these variables was verified. The external verification in the cohort shows good discrimination and calibration power especially our high C-index in the validation cohort indicates that due to its large sample size, this nomogram can be widely and accurately applied."
These results are still very much overstated, especially with respect to the nomogram. Sample size isn't that large (it is for an MDR-cohort, which tend to be very small, but it's still on the small end for predictive modelling)

* Where does the "compliance" come in for this study?

* "In our study, thrombocytopenia was identified as a high-risk factor for the development of multidrug-resistant pulmonary tuberculosis, which is consistent with previous research findings."
It is always implied that this study is about patients presenting with MDR-TB (vs. non-MDR). However, the rest of this paragraph refers to thrombocytopenia being a risk factor for side-effects and death in TB treatment, implying that if this is in line with this research, that these are patients with a history of TB or TB treatment failure. If so, the history of TB (treatment failure) itself is likely the main predictor and thrombocytopenia is just a proxy for this. The authors should really specify the TB history of the involved patients in this study.

It's also unclear why the section on previous TB treatment was deleted.

The methodological limitations mentioned in previous reviews are still not addressed in the manuscript (e.g., in the limitations section of the discussion) with the exception of some cosmetic changes in the conclusion.

Additional comments

Dear authors,

Would it be possible to have the manuscript be proofread by a native English speaker with MDR-TB knowledge?
My experience is more in the methodological area, and whenever I read your manuscript again and try to understand the clinical parts, I find inconsistencies and things that are unclear. I don't think either of us wants to keep going through ever more cycles of review, so it would be good for you to have a colleague proofread it for you and give suggestions.

To reiterate, the set-up of the study itself is good and prediction for MDR-TB is very valuable, but the manuscript as a whole still doesn't adequately deliver on this, with at least part of the problem being imprecise language that can mislead the readers.

---

## Round 0.4 · Minor Revisions

Please respond to ALL the comments of the reviewer. As noted by the reviewer, several of the points they raise have been raised several times already. If you are unable or unwilling to engage with these comments then we may be forced to reject your submission.

Reviewer 2 ·

Basic reporting

The professional language of the manuscript has greatly improved and has reached an acceptable level for publication. There are still a few issues to be resolved, but as far as I'm concerned, a necessary major step towards publication has been taken.

Please find below a find a few remaining comments where the content is imprecise and should be updated.

Line 118: "External validation was performed using the bootstrapping validation method."
I assume that bootstrapping was used for the standard error / variation, as the probabilistic predictions itself can be directly obtained from the validation cohort?

Line 155: "The eight differential factors selected using the LASSO method closely corresponded to the eight clinical features tentatively hypothesized in Figure 1."
Please delete this sentence. Several versions of this have been suggested in the manuscript, and they are all irrelevant. It is the same 8 variables, obviously, because these were the 8 you included in the first place. (It's like saying: "we locked 8 people in a room and when we opened the door again later, remarkably, the 8 people in the room closely corresponded to the people we put in the room!") The only thing relevant mentioning is that they all remained non-zero, which the previous 2 sentences and the next sentence already cover, forming a consistent message:
"Figure 2 demonstrates the optimal lambda value and shows the preservation of 8 significant variables at this value. These significant variables had non-zero coefficients in the LASSO regression model. This indicated that these eight clinical features might help evaluate MDR-TB."

Figure 2 (left) still has issues (see validity section for details)

Experimental design

Overall, the manuscript now contains all the necessary information.

The only part that is unclear to me (and only partially covered by the limitations) is the patient mix.
Specifically:
* It is not clear when and how the outcome (MDR or not) was determined. Ideally, the authors specify the resistance profile testing procedure in the methods.
* It is not clear if patients are new patients, patients switching treatment, patients returning after a relapse and/or treatment failure, etc. The high percentage of MDR suggests a mix?
This should ideally be reported.

Additionally, a distinction should be made between a diagnostic algorithm (which this one supposedly is) and a prognostic algorithm (predicting future / acquired MDR-TB). A few hints towards the latter are present (and one explicit, on line 235) but this is likely incorrect.
E.g., finding thrombocytopenia as a predictor in your study likely implies that it may be a diagnostic indicator (associated with current (possibly undiagnosed) MDR-TB) rather than a prognostic indicator (predictive for future MDR-TB).

Validity of the findings

Line 165: "and 0.825 for the validation cohort (Figure 3)", and also 0.825 in Figure 3 - this doesn't match the 0.808 of the abstract and line 183.

Some of the predictors are confounded with, and possibly proxies for severity of the disease in general and history of TB. This isn't a problem as the manuscript is about prediction modelling, not finding associations, but this should be mentioned as a limitation in the discussion.

As mentioned in previous reviews, there are some issues with disease duration.
Additionally, excluding a variable based on low effect size is not good practice because this is scale dependent. Both variables that you remove (disease duration and PLT) have very large values which naturally results in a small effect size (as effect size is the effect associated with a difference of "1 unit", with one unit being a very small difference for those two variables).
Additionally, again referring to another previous review: Figure 2 on the left is still very hard to interpret. This is partially due to scale differences as well as some parameters start very close to 0 while others are far from 0, but it also misses readable labels that indicate which line is which variable.

Additional comments

Thank you for the updated manuscript. This version is much improved from the previous one. If the editor follows my suggestions/advice, please do consider and address the remaining comments, some of which have been repeated multiple times in previous reviews.

---

## Round 0.5 · accepted · Accept

The authors have addressed all the reviewer' comments and, I am happy with the current version of the manuscript.